# Intraarticular Injections of Mesenchymal Stem Cells in Knee Osteoarthritis: A Review of Their Current Molecular Mechanisms of Action and Their Efficacy

**DOI:** 10.3390/ijms232314953

**Published:** 2022-11-29

**Authors:** Emérito Carlos Rodríguez-Merchán

**Affiliations:** 1Department of Orthopedic Surgery, La Paz University Hospital, Paseo de la Castellana 261, 28046 Madrid, Spain; ecrmerchan@hotmail.com; 2Osteoarticular Surgery Research, Hospital La Paz Institute for Health Research—IdiPAZ (La Paz University Hospital—Autonomous University of Madrid), 28046 Madrid, Spain

**Keywords:** MSCs, knee, osteoarthritis, mechanisms of action, efficacy

## Abstract

More than 10% of the world’s population suffers from osteoarthritis (OA) of the knee, with a lifetime risk of 45%. Current treatments for knee OA pain are as follows: weight control; oral pharmacological treatment (non-steroidal anti-inflammatory drugs, paracetamol, opioids); mechanical aids (crutches, walkers, braces, orthotics); therapeutic physical exercise; and intraarticular injections of corticosteroids, hyaluronic acid, and platelet-rich plasma (PRP). The problem is that such treatments usually relieve joint pain for only a short period of time. With respect to intraarticular injections, corticosteroids relieve pain for several weeks, while hyaluronic acid and PRP relieve pain for several months. When the above treatments fail to control knee pain, total knee arthroplasty (TKA) is usually indicated; however, although a very effective surgical technique, it can be associated with medical and postoperative (surgery-related) complications. Therefore, it seems essential to look for safe and effective alternative treatments to TKA. Recently, there has been much research on intraarticular injections of mesenchymal stem cells (MSCs) for the management of OA of the knee joint. This article reviews the latest information on the molecular mechanisms of action of MSCs and their potential therapeutic benefit in clinical practice in patients with painful knee OA. Although most recent publications claim that intraarticular injections of MSCs relieve joint pain in the short term, their efficacy remains controversial given that the existing scientific information on MSCs is indecisive. Before recommending intraarticular MSCs injections routinely in patients with painful knee OA, more studies comparing MSCs with placebo are needed. Furthermore, a standard protocol for intraarticular injections of MSCs in knee OA is needed.

## 1. Introduction

Primary knee osteoarthritis (OA) is a degenerative disease that causes severe joint pain, which is not easy to control, in patients who suffer from it. The increasing average life expectancy of the world’s population and the increasing prevalence of obesity make knee OA a growing economic burden on healthcare systems worldwide [1,2]. More than 10% of the world’s population has knee OA, with a lifetime risk of knee OA being 45% [3]. 

Current treatments for pain associated with OA include weight control; oral pharmacological treatment (non-steroidal anti-inflammatory drugs, paracetamol, opioids); mechanical aids (crutches, walkers, braces, orthotics); therapeutic physical exercise; and intraarticular injections of corticosteroids, hyaluronic acid, and platelet-rich plasma (PRP) [4,5]. However, these treatments tend to relieve joint pain only in the short term—several weeks for corticosteroids and several months for hyaluronic acid and PRP [6,7].

Given that knee OA is a chronic disease, long-term treatment typically requires a total knee arthroplasty (TKA). Although it is an effective surgical technique, it is not free from complications. Therefore, patients with knee OA whose pain cannot be controlled with the previously mentioned conservative and non-aggressive measures continue to seek therapies that are not as aggressive as TKA, such as intraarticular injections of mesenchy-mal stem cells (MSCs) [1,8,9].

MSCs are bone marrow-populating cells that are different from hematopoietic stem cells, which have a broad proliferative potential and the capacity to differentiate into various cell types, including adipocytes, cardiomyocytes, chondrocytes, myocytes, neurons, and osteocytes. MSCs are essential for maintaining bone marrow homeostasis and controlling the maturation of both hematopoietic and non-hematopoietic cells. Although the cells are characterized by the expression of many surface antigens, none appear to be solely expressed on MSCs. Apart from bone marrow, MSCs are also located in other tissues, such as adipose tissue, cord blood, liver and fetal tissues, and peripheral blood [10].

According to Jang et al., embryonic stem cells and induced pluripotent stem cells are transformed into chondrocytes or MSCs; thus, they can be administered by injection into the joint cavity in patients with knee OA. MSCs are also known to have immunomodulatory properties, e.g., the potential to enhance cartilage recovery and restore knee health [11].

The aim of this article is to review MSCs’ molecular mechanisms of action and the efficacy of their intraarticular injection in patients with painful knee OA.

## 2. MSCs’ Molecular Mechanisms of Action

Results of a PubMed (MEDLINE) search of studies related to MSCs in knee OA were analyzed. The searches were from the beginning of the search engine until 31 October 2022 using the keywords “Knee osteoarthritis MSCs”. Only the studies on MSCs in knee OA that the author considered to be of most interest were included. PubMed found 391 articles, of which 77 were selected. Those that seemed most directly related to the title of this article were chosen, i.e., 77 articles.

MSCs play significant roles in the repair and regeneration process (Figure 1). These include the reduction of cell death to continually replace lost cells, the secretion of trophic factors that stabilize the extracellular matrix, and the suppression of immune cell activation to prevent inflammation [12].

In one study, exosomes were harvested from ESC-MSCs in conditioned culture media by a sequential centrifugation process. Then, ESC-MSCs or their exosomes were intraarticularly injected. An in vitro model with primary mouse chondrocytes stimulated with interleukin-1 beta (IL-1β) was used to assess the impact of the conditioned medium with or without exosomes and titrated doses of isolated exosomes for two days, before immunocytochemistry or western blot analysis. Destabilization of the medial meniscus (DMM) surgery on knees of C57BL/6 J mice was used as an OA model [13]. The results revealed that exosomes from human embryonic stem cell-induced MSCs (ESC-MSCs) had a beneficial therapeutic impact against OA by balancing the production and decomposition of the extracellular matrix (ECM) of chondrocytes [13].

Using a Sprague–Dawley rat design of collagenase II and IL-1β-induced OA chondrocytes, long non-coding ribonucleic acid malat-1 from human MSC (hMSC)-delivered extracellular vesicles (EVs) promoted chondrocyte proliferation, alleviated chondrocyte inflammation and cartilage degeneration, and enhanced chondrocyte repair [14].

A study by Yang et al. used tropoelastin as the injection medium and compared it with classic media, such as normal saline, hyaluronic acid, and platelet-rich plasma (PRP), in intraarticular MSC injection. The authors found that tropoelastin promoted the emigration of infrapatellar fat pad MSCs (IPFP-MSCs) and protected knee cartilage from OA damage by enhancing cell adhesion and activating the integrin beta-1/extracellular signal-regulated protein kinase 1/2/vinculin pathway. These findings provided new insights into intraarticular injections of MSCs for the treatment of OA [15].

In one study, cultured MSCs originating from three types of tissues (bone marrow, adipose tissue, and synovium) were treated with IL-1β and tumor necrosis factor-alpha or not to attain conditioned media. Each conditioned medium was used to analyze paracrine factors related to cartilage restoration by liquid chromatography-tandem mass spectrometry. MSCs from these tissues expressed 93 proteins under normal circumstances and 105 proteins under inflammatory circumstances [16]. The differentially expressed proteins might contribute to the regeneration of damaged cartilage.

In a suspended synovium culture model, MSCs were released from the synovial membrane across a medium into a non-contacting culture dish [17]. The authors found that, in knees with OA, endogenous MSCs were possibly similarly mobilized from the synovium through the synovial fluid, acting in a protective manner. In the natural course of OA, however, the number of mobilized MSCs is limited, leading to OA progression. In a rat OA model, it was also noticed that injections of synovial MSCs inhibited the progression of cartilage degeneration. The injected synovial MSCs emigrated into the synovial membrane, maintained their MSC characteristics, and augmented gene expressions of TSG-6, PRG-4, and bone morphogenetic protein-2. That is, exogenous synovial MSCs could facilitate anti-inflammation, lubrication, and formation of cartilage matrix in osteoarthritic knees [17].

It has been shown that chronic swelling results in excess Ca^2+^ transfer from the endoplasmic reticulum to the mitochondria, leading to mitochondrial calcium overload and further mitochondrial harm [18]. Moreover, under conditions of chronic inflammation, injured mitochondria accumulate over time in MSCs due to mitophagy by activation of the Wnt/β-catenin pathway, which impairs MSC differentiation. Zhai et al. isolated tissue-specific MSCs in the periodontal ligament, termed periodontal ligament stem cells, from healthy patients and patients with periodontitis. Based on the mechanistic invention, intracellular microenvironment (esterase and low pH)-responsive nanoparticles were constructed to capture Ca^2+^ surrounding mitochondria in MSCs to control MSC mitochondrial calcium flux against mitochondrial dysfunction. The nanoparticles were able to liberate siRNA from MSCs to restrain the Wnt/β-catenin pathway and control mitophagy of the initially dysfunctional mitochondria. The aforementioned nanoparticles (“nanorepairers”) physiologically reestablished the activity of mitochondria and MSCs, which could be a new effective therapy against OA [18]. Figure 2 summarizes the cellular effect of the MSCs’ action.

## 3. Efficacy of Intraarticular MSCs Injections in Knee Osteoarthritis

### 3.1. Experimental Studies

A systematic review of animal studies concluded that intraarticular injections of MSCs could not be recommended for knee OA clinical trials. They also stated that, based on the internal and external validity of animal studies, high-quality experimental studies and greater efforts to translate preclinical studies to clinical trials were required at that time (in 2018) [19].

The safety and efficacy of intraarticular allogenic MSC injections was investigated in a pig OA experiment after bilateral medial knee meniscectomy [20]. Bone marrow-originated MDCs (BM-MSCs) were labelled with superparamagnetic iron oxide (SPIO) nanoparticles to permit cell tracing by magnetic resonance imaging (MRI). At a concentration of ≤20 µg/mL, SPIO nanoparticles were not toxic to BM-MSCs. Four weeks following surgery, OA lesions were noticed on the MRI. Between 8 and 24 h after the injections, the SPIO-marked BM-MSCs were displaced into the damaged portion of the cartilage. In addition, histological and immunohistochemistry analysis found no significant difference between the right knee (treatment group) and the left knee (control group). The appropriate concentration of SPIO nanoparticles for labelling BM-MSCs was 20 µg/mL, while allogenic MSCs were able to move into the impaired cartilage and accumulate around it. No significant differences were found between the treatment group and the control group [20].

One study compared the efficacy of extracorporeal shockwave therapy (ESWT), Wharton’s jelly MSCs (WJ-MSCs), and a combination of ESWT and WJ-MSCs in early knee OA in rats. The results of the pathological study, micro-computed tomography, and immunohistochemistry stain demonstrated that all three options significantly improved early knee OA. The combined therapy group augmented the bone volume and trabecular thickness, as well as diminished the synovitis more than ESWT or WJ-MSCs alone. However, there was no significant difference in the combined ESWT and WJ-MSCs, as demonstrated in the expressions of IGF-1 and transforming growth factor (TGF)-β1, and the decrease in the TUNEL activity in OA knees. Moreover, WJ-MSC therapy significantly augmented the expression of type II collagen compared with ESWT and ESWT combined with WJ-MSCs in OA knees. In the mechanistic factors analysis, a synergistic effect was observed by ESWT combined with WJ-MSCs in the expression of RUNX-2, SOX-9, and collagen Xα1 on knee OA [21].

It has been observed that IPFP-MSC-derived exosomes protected articular cartilaginous tissue from injury and improved gait anomaly in OA mice while keeping cartilage homeostasis, a mechanism that could be related to miR100-5p-regulated constraint of the mTOR-autophagy pathway. Wu et al. investigated the role and underlying mechanisms of IPFP-MSC-derived exosomes on OA in vitro and in vivo. At that time (2019), those authors stated that IPFP-MSC-derived exosomes could have utility in the treatment of knee OA, given that it was relatively easy in clinical practice to obtain human IPFP from patients with OA by arthroscopic surgery [22].

One study found that intraarticular injection of human umbilical cord MSCs expressing miR-140-5p induced cartilage self-repair in rat OA, highlighting the potential therapeutic utility of such injections in OA treatment [23]. In another study of adult male albino rats, it was observed that intraarticularly-injected umbilical cord blood MSCs cured knee OA better than when they were intravenously injected [24].

In a rat OA model, it was found that selective administration of kartogenin to synovial fluid-derived MSCs (SF-MSCs) by engineered exosomes produced a uniform dissemination of kartogenin in the cytosol, increased its effective concentration in the cell, and strongly promoted chondrogenesis of SF-MSCs in vitro and in vivo [25]. Using a rat OA model, other authors observed that exosomes from human bone marrow MSCs (BM-MSCs) had a beneficial therapeutic impact on OA by diminishing senescence and the death of chondrocytes. This result suggested that MSC-derived exosomes could have therapeutic value in OA [26].

In a rat knee OA model, it was demonstrated that low-intensity pulsed ultrasound (LIPUS) improved the therapeutic effectiveness of MSCs in cartilage reconstruction by increased autophagy-mediated exosome liberation [27]. In MSCs isolated from rat bone marrow in vitro, results showed that LIPUS facilitated exosome liberation from MSCs by triggering autophagy. The in vivo results showed that LIPUS substantially potentiated the positive impact of MSCs in OA cartilaginous tissue. This impact was substantially reduced by GW4869, an inhibitor of exosome liberation [27].

In a murine OA model, mouse ADSCs were acquired from adipose tissue and transfected with modified RNA. The results of histological and immunohistochemical analyses of knee joints harvested at 4 and 8 weeks after OA induction indicated that insulin-like growth factor 1 (IGF-1)-adipose derived stem cells (ADSCs) had a better therapeutic effect than native ADSCs. This outcome was shown by an inferior histological Osteoarthritis Research Society International score and less ECM loss. Such results supported the potential therapeutic utility of IGF-1-ADSCs for the treatment of OA and cartilage repair in clinical practice [28].

In one study, articular cartilage defects were created in the intertrochlear groove of articular cartilage in rabbit femurs. Integrin α10-MSCs were labeled with SPIO nanoparticles co-conjugated with rhodamine B to allow visualization by both MRI and fluorescence microscopy. The results showed emigration and homing of human integrin α10β1-selected MSCs to cartilage defects in the rabbit knees following intraarticular administration, as well as chondrogenic differentiation of MSCs in regenerated cartilage tissue [29]. 

A recent publication evaluated the effectiveness of BM-MSC management in cartilage repair, utilizing a rat experiment of monoiodoacetate-induced AO of the knee joint. OA was induced in the knee joint of rats by an intracapsular injection of monoiodoacetate (2 mg/50 *μ*L) on day zero. The authors concluded that BM-MSCs could be an effective treatment for inflamed knees, and that their effect could be mediated by their anti-inflammatory and antioxidant potential [30].

Ai et al. found that MSCs and MSC-EVs reduced OA pain through direct action on peripheral sensory neurons [31]. In their study, the authors elicited knee OA in adult male C57BL/6J mice by DMM surgery. The DMM mice treated with MSCs and MSC-EVs did not show the pain-related behavioral changes (i.e., locomotion, digging, and sleep) that the untreated DMM mice did. The lack of pain-related behaviors in the MSC/MSC-EV-treated mice was not due to diminished joint damage, but rather to the knee-innervating sensory neuron hyperexcitability that was observed in the untreated DMM mice. Moreover, they found that NGF-induced sensory neuron hyperexcitability was averted with MSC-EV management [31].

In a medial meniscal transection pre-clinical model of OA, sodium alginate microencapsulation of human MSCs modulated the paracrine signaling response and improved the efficacy of OA treatment. Three weeks post-surgery, after OA was established, intraarticular injections of encapsulated hMSCs or nonencapsulated hMSCs were administered. Six weeks post-surgery, microstructural changes in the knee joint were quantified with micro-computed tomography. Encapsulated hMSCs diminished articular cartilage degeneration and subchondral bone remodeling [32]. 

In a systematic review of animal models and cell doses, it was observed that rats were the most frequently employed species for modeling knee OA, and that anterior cruciate ligament transection was the most frequent approach used for producing OA [33]. A correlation was found between the cell doses and the body weight of the animals. In clinical trials, there was a great disparity in the dose of MSCs used to manage knee OA, ranging from 1 × 106 to 200 × 106 cells, with a mean of 37.91 × 106 cells. It was also found that in preclinical and clinical studies on knee OA, MSCs have significant potential for pain relief and tissue protection [33].

### 3.2. Clinical Studies

#### 3.2.1. Systematic Reviews

Table 1 summarizes systematic reviews and meta-analyses on the clinical efficacy of intraarticular injections of MSCs in knee OA [19,34,35,36,37,38,39,40,41,42,43,44,45,46,47,48,49,50,51,52,53,54,55].

#### 3.2.2. Randomized Controlled Trials (RCTs)

Table 2 summarizes RCTs on the clinical efficacy of intraarticular injections of MSCs in knee OA [56,57,58,59,60,61,62].

## 4. Discussion

The protocol of an ongoing randomized placebo-controlled trial (the SCUlpTOR trial) aims to evaluate the efficacy and cost-effectiveness of intraarticular MSC injections with respect to relieving joint pain and achieving structural improvement in people with tibiofemoral knee OA (trial registration numbers: Australian New Zealand Clinical Trials Registry [ACTRN1262620000870954]; U1111-1234-4897). Future results of this trial will help considerably in determining the efficacy and cost-effectiveness of intraarticular MSC injections in knee OA [63].

The goal of MSC treatment in knee OA is to be holistic, with the aim of achieving restoration of all impaired articular components. The paracrine impact of the MSCs’ secretome is fundamental for the regenerative ability of these cells. Triggering of local knee-joint-specific MSCs produces an immunomodulatory, anti-catabolic, anti-apoptotic, and chondrogenic stimulus [64]. 

Preclinical studies have shown the symptom- and illness-modifying impact of MSC treatment. In clinical practice, there is proof that autologous and allogeneic MSC treatment leads to a substantial improvement in symptoms and functional results. However, there are conflicting clinical results in the literature. Although MSC therapy has produced promising results, its efficacy is still unclear. The variety of cell origins, isolation, culture methods, and other circumstances make it difficult to compare studies. The clinical translation of their illness-modifying impact has not yet been achieved [64].

A molecular tool that could predict the impact of the osteoarthritic joint microenvironment on cartilage repair has recently been investigated [65]. For this purpose, 6 different promoters (hIL6, hIL8, hADAMTS5, hWISP1, hMMP13, and hADAM28) were generated in a 3-dimensional pellet culture model and stimulated with OA synovium-conditioned medium (OAs-cm) attained from 32 patients with knee OA. Cartilage formation was evaluated histologically and by quantification of sulfated glycosaminoglycan formation. The authors proved that OAs-cm from various patients had significantly different effects on sulfated glycosaminoglycan formation. Furthermore, they observed substantial correlations between the impact of OAs-cm and cartilage production and promoter reporter results. The predictive usefulness of measuring 2 promoter reporters with an independent group of OAs-cms was confirmed. This new tool was able to predict the impact of 87.5% of the OAs-cm joint microenvironment on MSC-based cartilage production. This is a relevant first step toward personalized cartilage repair approaches for patients with OA, enabling the prediction of whether the OA joint microenvironment is permissive for cartilage repair; thus, it could be of great importance in determining the success of cartilage repair strategies using MSCs [65].

There are some major factors that could impact the efficacy of intraarticular MSC injections, such as allogenic versus autologous cells, primary cells versus cultured cells, differentiated versus undifferentiated cells, licensed versus unlicensed cells, the variation of cell preparations, and the clinical conditions of the recipients.

With respect to allogenic versus autologous MSCs, the possible hazards and restraints of using autologous versus allogeneic MSCs in clinical practice are still being argued, such as the possible influence of donor–donor heterogeneity. Figure 3 shows the reported advantages and disadvantages of allogeneic and autologous MSCs in preclinical and clinical practice [66].

Regarding primary MSCs versus cultured cells, native bone marrow extracellular matrix renders a unique microenvironment that reduces the growth of MSCs in serum-free media and maintains MSC quality in terms of replication, differentiation, and bone morphogenetic protein-2 responsiveness. The use of a potent culture system consisting of native tissue-specific ECM and defined serum-free media will permit us to prepare substantial amounts of MSCs while simultaneously maintaining their stem cell characteristics for cell-based treatments [67].

In terms of differentiated MSCs versus undifferentiated MSCs, a study compared the outcomes of grafting into the rat contused spinal cord undifferentiated ADSCs versus ADSCs induced by two different methods to create differentiated nervous tissue. The findings of this study suggested that ADSCs were able to differentiate into neural-like cells in vitro and in vivo. Neural differentiated ADSCs, however, did not lead to better functional recovery than undifferentiated ones [68].

With regard to licensed MSCs versus unlicensed MSCs, in vitro licensing before therapeutic application could lead to a more foreseeable immunomodulatory and reparative reaction to MSC treatment compared with in vivo inflammatory licensing by the recipient environment [69]. Some authors rendered strong evidence for the use of TGF-β1 licensing as an unconventional approach for improving MSC immunosuppressive ability [70]. In 2021, Lu and Qiao stated that, despite the heterogeneity, pre-licensing did not impact the cell cycle and stemness of human bone marrow-derived MSCs. The osteogenic potency was reduced and the chondrogenic potency was augmented, while the adipogenic potency was stable in licensed MSCs. Licensed MSCs also demonstrated more efficacious immunomodulatory ability, including expression of related chemokines, cytokines, surface molecules, and receptors [71].

With respect to the variation in cell preparation, exosomes liberated by MSCs have been found to be good candidates for cartilage injuries and OA management, and exosomes for clinical practice needed large-scale production. To this end, human synovial fluid MSCs were grown on microcarrier beads and then cultured in a dynamic 3-dimensional culture system. This culturing system successfully attained large-scale exosomes from synovial fluid MSC culture supernatants, indicating that this technique can generate a great amount of good manufacturing practice-grade exosomes. These exosomes could be used in exosome biology research and clinical OA management [72]. In one study, injectable ADSC-embedded alginate-gelatin microspheres were prepared by electrospray. Compared with traditional alginate microspheres, its support for ADSCs was better and demonstrated a better repair effect. This approach could be useful for cartilage tissue regeneration [73].

Regarding the clinical conditions of the recipients, it has been reported that 40 × 10^6^ MSCs were the most likely to accomplish optimal responses in patients with grade ≥2 knee OA. Although substantial ameliorations were found when using inferior (24 × 10^6^) and superior (100 × 10^6^) cell numbers, they caused persistent pain and inflammation [12].

The efficacy of intraarticular MSC injections for knee OA remains controversial, although most recent publications show short-term pain relief. Orthopedic surgeons managing patients with knee OA are increasingly interested in MSCs, even though clinical information and basic scientific data are indecisive. More research comparing MSCs with placebo is required.

A standard protocol for intraarticular MSC injections in knee OA is needed. This protocol must include the following: cell selection, authentication (phenotypic analysis and multipotent differentiation potential, particularly differentiation with progenitor cells), culture or expansion techniques, dosages, and rehabilitation program after treatment [74]. 

The main limitation of this review is that the selection of articles that were ultimately analyzed was subjective, i.e., those that we deemed most directly related to the title of the article. Thus, it is feasible that some important articles were not included. This article is not a systematic review of the literature, but a narrative review of the articles we found most relevant.

## 5. Conclusions

There has recently been much investigation on intraarticular MSC injections for the management of OA of the knee joint. Although the majority of recent reports claim that intraarticular injections of MSCs alleviate knee pain in the short term, their effectiveness remains controversial, given that current scientific data on MSCs is indecisive. Before advising intraarticular MSC injections routinely in patients with painful knee OA, more research comparing MSCs with placebo is required, as well as a standard protocol for intraarticular injections of MSCs in knee OA.

## Figures and Tables

**Figure 1 ijms-23-14953-f001:**
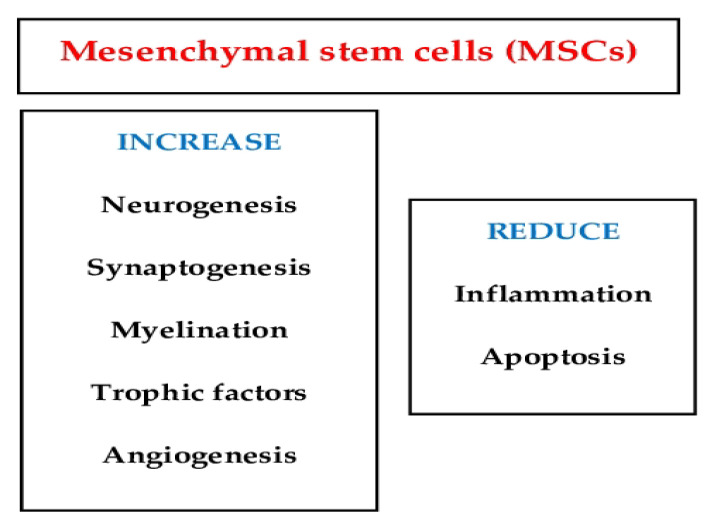
This image shows that the function of mesenchymal stem cells plays significant roles in the repair and regeneration process.

**Figure 2 ijms-23-14953-f002:**
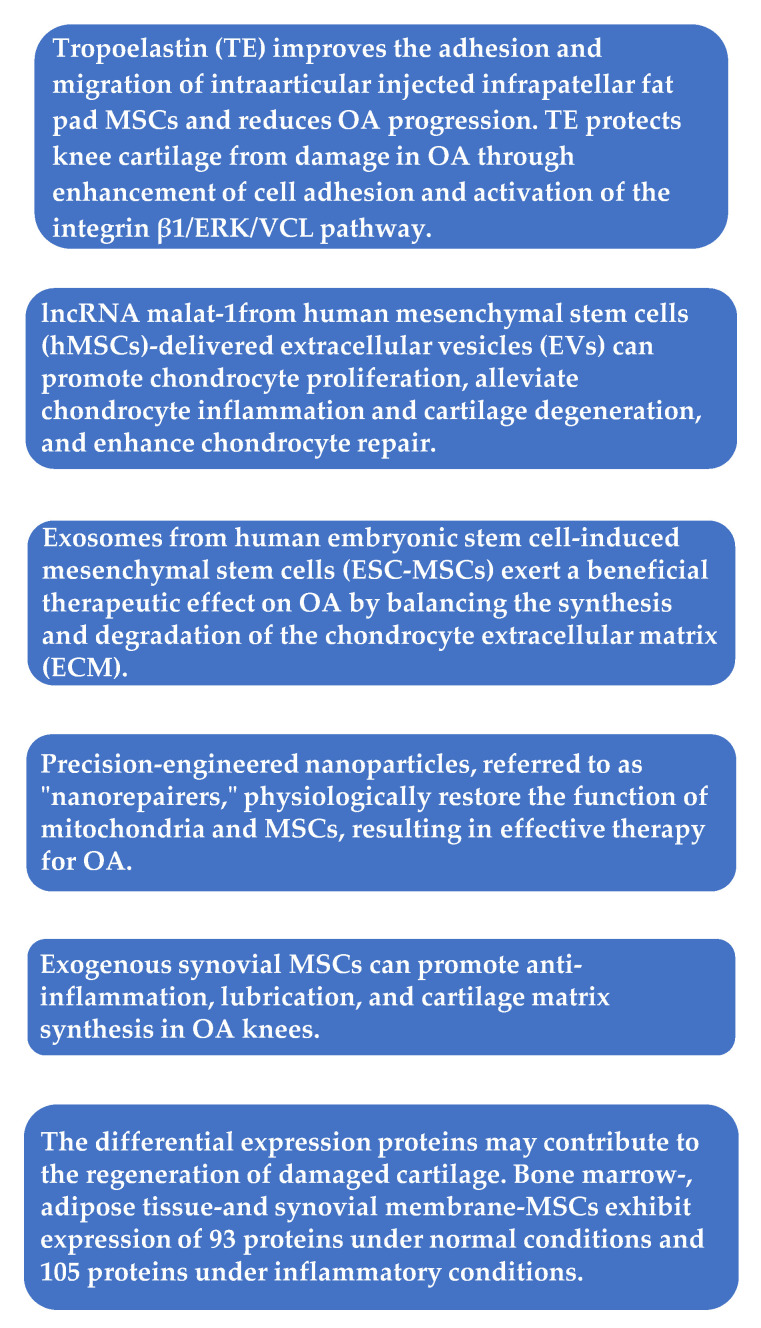
Mesenchymal stem cells (MSCs): cellular effect of their action. OA = osteoarthritis; ERK = extracellular signal-regulated protein kinase; VCL = vinculin; lncRNA = long non-coding RNA.

**Figure 3 ijms-23-14953-f003:**
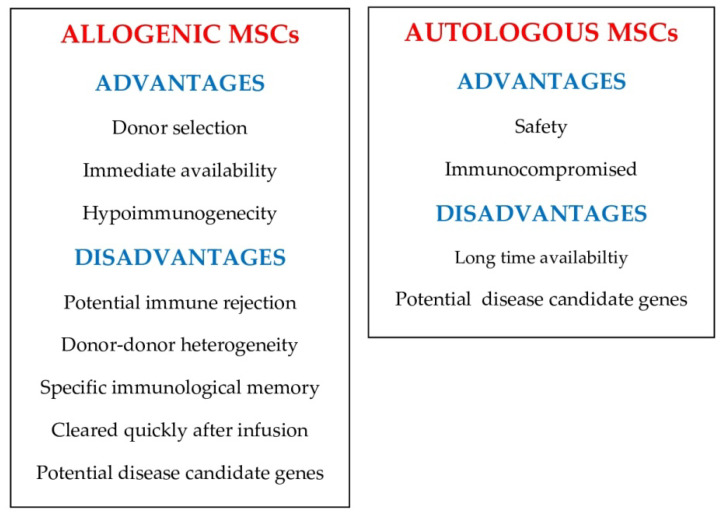
Pros and cons of allogenic and autologous mesecnchymal stem cells (MSCs).

**Table 1 ijms-23-14953-t001:** Systematic reviews and meta-analyses on the effectiveness of intraarticular injections of MSCs in OA of the knee joint.

Authors [Reference]	Year	Methods and Results	Participants	Level of Evidence	Conclusions
Xia et al. [34]	2015	MSCs injection had no substantial impact on pain.	314 patients (7 RCTs)	I	The results of this review should be validated utilizing methodologically precise trials.
Cui et al. [35]	2016	MSC therapy in subjects with knee OA demonstrated continual effectiveness for 2 years.	565 patients (18 clinical trials)	I	MSC utilization improved the overall results.
Xing et al. [19]	2018	This study demonstrated that modest reliance could be placed on safety of MSCs treatment for knee OA.	23 animal studies	I	More high-quality research with high internal and external validity is still needed.
Ha et al. [36]	2019	All reports except two found significantly superior clinical results in the MSCs group.	17 studies in patients with knee OA (6 RCTs, 8 prospective observational studies, 3 retrospective case-control studies).	III	Intraarticular MSCs render ameliorations in pain and function in knee OA at short-run follow-up (<28 months) in many cases.
Kim et al. [37]	2019	This study found significant improvements after treatment.	220 patients (5 RCTs)	II	Intraarticular MSCs have limited evidence in pain alleviation and functional betterment in knee OA.
Di Matteo et al. [38]	2019	Twenty-three manuscripts were included in the final analysis.	23 manuscripts about patients with knee OA (only 4 were RCTs)	NA	The poor quality of the reported studies averted any recommendation on the utilization of either product in a clinical practice.
D’Arrigo et al. [39]	2019	Encouraging in vitro outcomes were obtained in terms of enhanced cell proliferation, decrease of swelling.	Twenty in vivo and in vitro studies were analyzed.	NA	The different effects of EVs and secretome, and the identification of subjects who may benefit more from intraarticular injections of MSCs must be clarified.
Álvarez Hernández et al. [40]	2020	Data demonstrated clinical amelioration in 60% of subjects. Structural benefit was found in 50% of subjects.	169 patients (3 RCTs, 6 QCTs)	NA	Intraarticular implants of MSCs appeared to be safe with no serious complications. Low-quality evidence averts conclusions regarding efficacy.
Song et al. [41]	2020	MSC therapy could substantially reduce VAS in a 1-year follow-up study compared with controls.	58 patients (15 RCTs, two retrospective studies and two cohort studies)	NA	These authors suggested that MSC treatment could be efficacious and safe therapy for the treatment of OA.
Dai et al. [42]	2021	Compared with placebo, there was no significant difference in VAS for pain, WOMAC pain score, WOMAC function score, or WOMAC stiffness score for MSCs.	13 RCTs (patients)	I	Intraarticular MSC injection was not encountered to be superior to placebo in pain alleviation and functional betterment for subjects with knee OA.
Maheshwer et al. [43]	2021	There was no substantial difference in pain alleviation between MSC treatment and controls.	439 patients (25 studies)	II	MSCs rendered functional benefit only in subjects who experienced concurrent surgery.
Qu et al. [44]	2021	MSC treatment substantially diminished VAS, WOMAC pain, WOMAC stiffness, and WOMAC function scores at a long-run follow-up (1 or 2 years).	476 patients (9 RCTs)	NA	The results of this study suggested that MSCs were a promising alternative for the management of subjects with knee OA.
Tan et al. [45]	2021	All studies reported amelioration in the results after MSC therapy.	440 knees (19 studies)	NA	Intraarticular injections of MSCs without any adjuvant therapies improved pain and function for OA.
Naja et al. [46]	2021	This study assessed 7 approaches with WOMAC at 1 year: injection of PRP, corticosteroids, MSCs, hyaluronic acid, ozone, administration of NSAIDs with or without the association of physiotherapy.	13 trials (patients)	NA	The results of treatments utilizing MSCs and PRP for the management of knee OA were associated with long-run improvements in pain and function.
Muthu et al. [47]	2021	At 6 months, culture expanded MSCs demonstrated pain alleviation.	767 patients (17 studies)	NA	Culture expansion of autologous MSCs was not a necessary factor to attain better results in the treatment of knee OA.
Zhao et al. [48]	2021	This meta-analysis compared AD-MSCs, LP-PRP, and placebo. At 6 months, VAS scores and WOMAC pain subscores demonstrated that AD-MSCs were the best treatment alternative.	43 studies (patients)	II	During 6 months of follow-up, AD-MSCs alleviated pain the best; LP-PRP was most efficacious for functional amelioration.
Jeyaraman et al. [49]	2021	At 6 months, 1 year and 2 years, AD-MSCs demonstrated substantially better VAS and WOMAC amelioration than BM-MSCs, respectively, compared to controls.	811 patients (9 studies)	NA	This study established the effectiveness, safety, and superiority of AD-MSC transplantation, compared to BM-MSC, in the treatment of OA.
Muthu et al. [50]	2021	These authors categorized the studies based on the MSC count used in them into four cohorts, namely <1 × 10^7^ MSCs (Cohort I), 1–5 × 10^7^ MSCs (Cohort II), 5–10 × 10^7^ MSCs (Cohort III), and >10 × 10^7^ MSCs (Cohort IV).	564 patients (14 studies)	NA	Cohort III demonstrated consistent substantial amelioration in pain and functional result analyzed compared to the other cohorts. Therefore, these authors advised a cell volume of 5–10 × 10^7^ cells.
Wei et al. [51]	2021	The MSCs were deemed superior over placebo for pain alleviation and ameliorated function in KOA, but demonstrated no substantial differences for cartilage regeneration. Among all the MSCs, the AD-MSCs most effectively alleviated pain.	203 patients (8 studies)	NA	The findings of this study suggested that MSCs were effective in the treatment of knee OA. However, the evidence did not support the utilization of MSCs for ameliorating cartilage repair in subjects with knee OA.
Wiggers et al. [52]	2021	After 1 year, 19 of 26 (73%) clinical outcome parameters ameliorated with MSCs compared with control.	408 patients (14 RCTs)	NA	These authors encountered a positive impact of autologous MSC therapy compared with control treatments on PROMs, and illness severity. The quality of this evidence was low.
Álvarez Hernández et al. [40]	2022	Clinical improvement was found in 60% of subjects. Structural benefit was seen in 50% of subjects.	169 patients (252 articles)	NA	Intraarticular implants of MSCs appeared to be safe, with no serious complications. Low-quality evidence precludes conclusions regarding effectiveness in this review.
Dhillon et al. [53]	2022	After a follow-up 23.4 months, weighted averages of the WOMAC, macroscopic ICRS, subjective IKDC, and VAS scores all demonstrated amelioration from before to after treatment.	385 patients (7 studies)	NA	Subjects experiencing management of knee OA with hUC-MSCs might be expected to improve.
Jeyaraman et al. [54]	2022	At 6 months, both direct and vehicle-based delivery of MSCs demonstrated substantially better VAS amelioration.	963 patients (21 studies)	NA	Employed methods of vehicle-based delivery of MSCs, such as PRP and hyaluronic acid, did not show better outcomes compared to direct delivery.
Shoukrie et al. [55]	2022	Substantial ameliorations were seen in the MSCs cohorts regarding KOOS, VAS, WOMAC, and MRI. Moreover, no serious complications were found.	10 studies (723 patients)	NA	Intraarticular injections of MSCs were efficacious and safe in alleviating pain and ameliorating motor function in subjects with knee OA in the short run.

MSCs = vesicles; QCTs = Qualifying Clinical Trials; VAS = Visual analog scale; WOMAC = Western Ontario and McMaster Universities Osteoarthritis Index; PRP = platelet-rich plasma; NSAIDs = nonsteroidal anti-inflammatory drugs; AD-MSCs = Adipose tissue-derived MSCs; LR-PRP = Leukocyte-rich platelet-rich plasma; BM-MSCs = Bone marrow-derived MSCs; PROMs = Mesenchymal stem cells; OA = Osteoarthritis; n = Number of patients; LoE = Level of evidence; RCTs = Randomized controlled trials; EVs = Extracellular Patient-related outcomes; ICRS = International Cartilage Regeneration and Joint Preservation Society; IKDC = International Knee Documentation Committee; hUC-MSCs = human umbilical cord-derived MSCs; KOOS = Knee injury and Osteoarthritis Outcome Score; MRI = Magnetic resonance imaging; NA = Not available.

**Table 2 ijms-23-14953-t002:** Randomized clinical trials (RCTs) on the effectiveness of intraarticular injections of MSCs in knee OA.

Authors [Reference]	Year	Methods and Results	Participants	Level of Evidence	Conclusions
Vega et al. [56]	2015	These authors randomized 30 subjects with chronic knee pain unresponsive to conservative management and exhibiting radiological evidence of OA into two cohorts of 15 subjects. The test cohort was treated with allogeneic bone marrow MSCs by intra-articular injection of 40 × 10(6) cells. The control cohort received intra-articular hyaluronic acid (60 mg, single dose).	30 patients	NA	Allogeneic MSC treatment might be a valid option for the treatment of chronic knee OA. The procedure was simple, did not need surgery, provided pain alleviation, and substantially ameliorated cartilage quality.
Lamo-Espinosa et al. [57]	2018	In this phase I/II multicenter randomized clinical trial with active control, no complications were found after autologous bone marrow-derived MSCs (BM-MSCs) administration or during the follow-up. BM-MSCs-administered subjects improved according to VAS at the end of follow up.	30 patients	NA	Single intraarticular injection of in vitro expanded autologous BM-MSCs was a safe and feasible technique that resulted in long-run clinical and functional amelioration of knee OA.
Matas et al. [58]	2019	Subjects with symptomatic knee OA were randomized to receive hyaluronic acid at baseline and 6 months (hyaluronic acid, n = 8), single-dose (20 × 10^6^) UC-MSC at baseline (MSC-1, n = 9), or repeated UC-MSC doses at baseline and 6 months (20 × 10^6^ × 2; MSC-2, n = 9).	26 patients	NA	In this phase I/II trial, repeated UC-MSC therapy was safe and better than the comparative group at 1-year follow-up.
Lee et al. [59]	2019	Single injection of AD-MSCs led to a substantial amelioration of the WOMAC score at 6 months. In the control group, there was no significant change in the WOMAC score at 6 months.	24 patients	NA	An intraarticular injection of autologous AD-MSCs rendered satisfactory functional amelioration and pain alleviation for subjects with knee OA without causing complications at 6-month follow-up.
Lamo-Espinosa et al. [60]	2020	These authors assessed the clinical impact of a dose of 100 × 10^6^ cultured autologous BM-MSCs in combination with PRP (PRGF^®^) as adjuvant. No complications were found after BM-MSC administration or during follow-up.	60 patients	NA	Treatment with BM-MSC associated with PRGF^®^ was shown to be a viable therapeutic option for OA of the knee, with clinical improvement at the end of follow-up.
Bastos et al. [61]	2020	This study compared the clinical and laboratory results of intraarticular injections of culture-expanded bone-derived MSCs with or without PRP to intraarticular corticosteroid injections for the management of knee OA.	47 patients	II	An intraarticular injection of bone marrow-derived culture-expanded MSCs with or without the addition of PRP was efficacious in ameliorating the diminishing function and symptoms caused by knee OA at 12-month follow-up.
Hernigou et al. [62]	2021	These authors compared subchondral bone to intraarticular injection of bone marrow concentrate MSCs in bilateral knee OA. The aim was to determine which one of them was better at postponing TKA at 15 years.	60 patients (120 knees)	NA	Implantation of MSCs in the subchondral bone of an osteoarthritic knee was more efficacious at delaying TKA than injection of the same intraarticular dose in the contralateral knee with the same degree of OA.

MSCs = Mesenchymal stem cells; OA = Osteoarthritis; n = Number of patients; LoE = Level of evidence; BM-MSCs = Bone marrow-derived MSCs; UC-MSCs = umbilical cord-derived MSCs; AD-MSCs = Adipose tissue-derived MSCs; PRP = Platelet-rich plasma; TKA = total knee arthroplasty; NA = Not available.

## Data Availability

Not applicable.

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
