# Peer review of "Intraarticular Injections of Mesenchymal Stem Cells in Knee Osteoarthritis: A Review of Their Current Molecular Mechanisms of Action and Their Efficacy"

_ijms, 2022, doi:10.3390/ijms232314953_

Round 1
Reviewer 1 Report
Intraarticular injections of mesenchymal stem cells in knee osteoarthritis: A review of their current molecular mechanisms of 3 action and their efficacy
This is an interesting subject. Rodriguez-Merchan went over 385 articles and cited 66 of them for an overview of the molecular mechanisms and clinical efficacy of MSC-based therapy for knee osteoarthritis. However, there are some major concerns:
For the first part, the review is superficial. The author did not describe Figures 1-3 well. They are quite simple, which is not necessarily directed to people actually working in this field and looking for new information.
For the second part, although it is a good approach to use Tables to summary clinical studies, some important information should be included. The additional categories should be: 1) patients including numbers (sample size), sex, age, and brief medical conditions (OA stage, medication, other diseases?); and 2) administration including types of MSCs, single or multiple injections, cultured or uncultured cells, and doses. The author should carefully use consistent terminology in Tables.
In Discussion, the author should mainly focus on discussion about potential factors that cause inconsistent clinical outcomes, and potential solutions. Otherwise, the reader cannot obtain a specific information from this review.
Author Response
REVIEWER-1
This is an interesting subject. Rodriguez-Merchan went over 385 articles and cited 66 of them for an overview of the molecular mechanisms and clinical efficacy of MSC-based therapy for knee osteoarthritis. However, there are some major concerns:
For the first part, the review is superficial. The author did not describe Figures 1-3 well. They are quite simple, which is not necessarily directed to people actually working in this field and looking for new information.
AUTHOR: I think that in figure 1 it is clear what MSCs increase and what they reduce. In figure 2, the cellular effect of their action has been summarized. Honestly. I think that in both figures the message is clear. Figure 3 has been deleted.
For the second part, although it is a good approach to use Tables to summary clinical studies, some important information should be included. The additional categories should be: 1) patients including numbers (sample size), sex, age, and brief medical conditions (OA stage, medication, other diseases?); and 2) administration including types of MSCs, single or multiple injections, cultured or uncultured cells, and doses. The author should carefully use consistent terminology in Tables.
AUTHOR: I have included two new categories in the tables (participants and level of evidence – IN BLUE). The rest of information can be found within the text. I think that too many columns in a table will make it very difficult to read.
In Discussion, the author should mainly focus on discussion about potential factors that cause inconsistent clinical outcomes, and potential solutions. Otherwise, the reader cannot obtain a specific information from this review.
AUTHOR: I have added a paragraph in the DISCUSSION and have changed the CONCLUSIONS to give a specific information to the potential readers of this article (SEE THE CHANGES IN RED):
DISCUSSION
The efficacy of intraarticular injections of MSCs for knee OA remains controversial, although most recent publications show short-term pain relief. Orthopedic surgeons managing patients with knee OA are more and more interested in MSCs, even though clinical information and basic scientific data are indecisive. More research comparing MSCs with placebo is required.
CONCLUSION
Lately, there has been much investigation on intraarticular injections of MSCs for the management of OA of the knee joint. Although the majority of recent reports claim that intraarticular injections of MSCs alleviate knee pain in the short run, their effectiveness endures controversial given that current scientific data on MSCs is indecisive. Before advising intraarticular MSCs injections routinely in subjects with painful knee OA, more research comparing MSCs with placebo is required.
Reviewer 2 Report
The main goal of the manuscript, that is to critically describe the mesynchemal stemm cells injections in osteoarthritis is very nice and ambitious. I find this story very positive and interesting and the attempts to identify, describe and discuss the effects of biological therapies in orthopeadics is definitely needed.
My main criticism concerns four issues:
1. The presentation of the results. I cannot find the figures as publication-quality at the moment. Although they contain some information, a huge amount of important data is missing here (e.g. reference, model organism used for the study or Author’s comment). Other than that, presentation of such important issues on figures created in word program using aggressive coloring is not very professional. Please re-think the idea of the figures and their presentation possibilities.
On the other hand, the tables are constructed more professionally, with a good proportion of the data. They are detailed enough. However, you could work two things:
· formatting the tables, e.g. align the text properly and unify font size and line spacing (in different columns they are different).
· Tables 2 and 3 are way to descriptive – please minimize the text, e.g. dividing some of the information from the third column in more specific columns (e.g. number of patients, level of evidence, etc.)
2. Many sentences are copied word-by-word from the original publications and therefore one might suspect plagiarism – please change the text accordingly.
3. Starting from the line 69 you start listing specific works and their results. This arrangement is suitable for a table rather than a survey article. In the review paper, it is necessary not only to list, but also to give a sequence (e.g. chronological or thematic) to the works cited. One must emerge from the other, and finally in science one discovery or observation mulls another. Merely describing the work done previously by others is not sufficient for a review article. Moreover, please specify in all paragraphs if the work was done on an in vitro model of OA, a cell-culture model, animal model on real patients. Maybe think of dividing the data presented here in such paragraphs.
4. In the whole text some of the statements are to strong: e.g.:
· line 81 “hMSCsmalat-1-EVs could be a potential new therapeutic option for patients with OA” – Pan et al. did not prove that, they used EVs from MSCc, since EVs are potent cargos of noncoding RNAs and studied malat influence on some of the processes in rat OA model
· „Intraarticular injection of mesenchymal stem cells (MSCs)…” – please be careful with such statements and specify the source of MSCs. I do not know any data so far with the use of isolated or culture-conditioned MSCs in human patients. Usually MSCs are the main component of the mixture injected. – please specify that
Some other issues:
5. Many abbreviations are not explained, e.g. β1/ERK/VCL pathway, IL-1β, TNF-α, siRNA lncRNA, IKDC, and others
6. The Introduction is short, but it presents the most important aspects of knee OA and the treatment possibilities. Maybe you could elaborate here more on other orthobiologics used for knee OA and present the form of injection. Moreover, putting here just one single reference to Jang et al. is neither sufficient nor necessary at this stage of the manuscript.
7. In paragraph 2 a short characteristics of MSCs, before coming to the details of their action is needed.
8. I do not understand what kind of key word were taken into account when Pubmed search was performed and what was the idea of this. Did you look for the mechanisms of action, mechanisms of action in OA or usage in knee OA? Please specify here. Moreover, what was the key to select 66 articles out of 385 fished out in the initial search?
9. In paragraph 2 you do not describe the molecular mechanisms of action of MSc, you present rather the cellular effect of their action. Therefore I think stating “A review of their current molecular mechanisms of 3 action and their efficacy” in the title is not accurate. You do not discuss any of the molecular mechanisms here.
10. Table 2 – please add information how many patients were included in particular studies
Minor changes:
11. Line 62 “The function of MSCs plays a significant role” – rather “of MSCs play a significant role..”
12. Line 85 “intraarticular injected infrapatellar fat” – rather “intraarticularly”
13. Line 90 “the differential expression proteins” – rather “the proteins which were differenatilly expressed”
14. Fig. 2 – “TE promotes performance”
15. Line 131 and 192 – “intra-articular” – rather “intraarticular”
16. Line 139. “by bilateral medial medial meniscectomy” – “after bilateral medial medial meniscectomy?”
17. Line 139 – “Super-139 paramagnetic iron oxide (SPIO) nanoparticles were labelled with bone marrow-derived 140 MDCs (BM-MSCs)” – rather cells were labelled with SPIOs, right?
18. Line 146. “right side (treatment group) and the left side (control group)” – what for these right and left sides refer to?
19. Line 153 “all three options significantly improved early knee OA” – please elaborate on that, what exactly was improved?
20. Line 196 – the explanation of the abbreviation BM-MSCs is not needed here, you should place it upwords
21. Table 2. “papers” -rather “manuscripts”
Author Response
REVIEWER-2
The main goal of the manuscript, that is to critically describe the mesynchemal stemm cells injections in osteoarthritis is very nice and ambitious. I find this story very positive and interesting and the attempts to identify, describe and discuss the effects of biological therapies in orthopedics is definitely needed.
My main criticism concerns four issues:
- The presentation of the results. I cannot find the figures as publication-quality at the moment. Although they contain some information, a huge amount of important data is missing here (e.g. reference, model organism used for the study or Author’s comment). Other than that, presentation of such important issues on figures created in word program using aggressive coloring is not very professional. Please re-think the idea of the figures and their presentation possibilities.
AUTHOR: I have changed the format of Figure 1 and amended the text of Figure 2 (Figure 3 has been deleted).
On the other hand, the tables are constructed more professionally, with a good proportion of the data. They are detailed enough. However, you could work two things:
- formatting the tables, e.g. align the text properly and unify font size and line spacing (in different columns they are different).
- Tables 2 and 3 are way to descriptive – please minimize the text, e.g. dividing some of the information from the third column in more specific columns (e.g. number of patients, level of evidence, etc.)
AUTHOR: I think that the Reviewers refers to Tables 1 and 2 (not 2 and 3), because there are only two tables in the manuscript. I have included two new columns in each table: PARTICIPANTS and LEVEL OF EVIDENCE.
- Many sentences are copied word-by-word from the original publications and therefore one might suspect plagiarism – please change the text accordingly.
AUTHOR: This has been changed.
- Starting from the line 69 you start listing specific works and their results. This arrangement is suitable for a table rather than a survey article. In the review paper, it is necessary not only to list, but also to give a sequence (e.g. chronological or thematic) to the works cited. One must emerge from the other, and finally in science one discovery or observation mulls another. Merely describing the work done previously by others is not sufficient for a review article. Moreover, please specify in all paragraphs if the work was done on an in vitro model of OA, a cell-culture model, animal model on real patients. Maybe think of dividing the data presented here in such paragraphs.
AUTHOR: In all paragraphs I have specified the type of model used.
- In the whole text some of the statements are to strong: e.g.:
- line 81 “hMSCsmalat-1-EVs could be a potential new therapeutic option for patients with OA” – Pan et al. did not prove that, they used EVs from MSCc, since EVs are potent cargos of noncoding RNAs and studied malat influence on some of the processes in rat OA model
AUTHOR: I have deleted this sentence.
- „Intraarticular injection of mesenchymal stem cells (MSCs)…” – please be careful with such statements and specify the source of MSCs. I do not know any data so far with the use of isolated or culture-conditioned MSCs in human patients. Usually MSCs are the main component of the mixture injected. – please specify that
AUTHOR: In all paragraphs I have specified the type of model used.
Some other issues:
- Many abbreviations are not explained, e.g. β1/ERK/VCL pathway, IL-1β, TNF-α, siRNA lncRNA, IKDC, and others
AUTHOR: The abbreviations have been expanded as shown below:
Integrin β1/ERK/VCL pathway
Integrin β1 = integrin beta-1
ERK1/2 = extracellular signal‑regulated protein kinase 1/2
Vinculin = VCL
IL-1β = Interleukin-1 beta
TNF-α = Tumor necrosis factor-alpha
siRNA = small interfering ribonucleic acid
lncRNA = Long non-coding ribonucleic acid
IKDC = International Knee Documentation Committee
- The Introduction is short, but it presents the most important aspects of knee OA and the treatment possibilities. Maybe you could elaborate here more on other orthobiologics used for knee OA and present the form of injection. Moreover, putting here just one single reference to Jang et al. is neither sufficient nor necessary at this stage of the manuscript.
AUTHOR: I think that in the initial version of the manuscript I already mentioned other intraarticular orthobiologics used for knee OA: hyaluronic acid and platelet-rich plasma.
- In paragraph 2 a short characteristics of MSCs, before coming to the details of their action is needed.
AUTHOR: I have included the following paragraph on the characteristics of MSCs:
Mesenchymal stem cells (MSCs) are bone marrow populating cells, different from hematopoietic stem cells, which have an broad proliferative potential and capacity to differentiate into various cell types, including adipocytes, cardiomyocytes, chondrocytes, myocytes, neurons, and osteocytes. MSCs are essential in maintaining bone marrow homeostasis and control the maturation of both hematopoietic and non-hematopoietic cells. The cells are characterized by the expression of many surface antigens, but none of them seems to be solely expressed on MSCs. Apart from bone marrow, MSCs are placed in other tissues, like adipose tissue, cord blood, liver and fetal tissues, and peripheral blood [BOVIS].
- I do not understand what kind of key word were taken into account when Pubmed search was performed and what was the idea of this. Did you look for the mechanisms of action, mechanisms of action in OA or usage in knee OA? Please specify here. Moreover, what was the key to select 66 articles out of 385 fished out in the initial search?
AUTHOR: The following paragraph explains how the search was done:
The searches were from the beginning of the search engine until 31 October 2022 using the following keywords: “Knee osteoarthriris MSCs”. Only the studies on MSCs in knee OA that the author considered to be of most interest were included. PubMed found 391 articles, of which 67 were selected. Those that seemed most directly related to the title of this article were chosen (67 articles).
- In paragraph 2 you do not describe the molecular mechanisms of action of MSc, you present rather the cellular effect of their action. Therefore I think stating “A review of their current molecular mechanisms of action and their efficacy” in the title is not accurate. You do not discuss any of the molecular mechanisms here.
AUTHOR: I have changed this as follows:
The aim of this article is to review the cellular effect of the action of MSCs and the efficacy of the intraarticular injections of MSCs in patients with painful knee OA.
- Table 2 – please add information how many patients were included in particular studies
AUTHOR: The number of patients (participants) has been included in Tables.
Minor changes:
- Line 62 “The function of MSCs plays a significant role” – rather “of MSCs play a significant role..”
AUTHOR: It has been changed.
- Line 85 “intraarticular injected infrapatellar fat” – rather “intraarticularly”
AUTHOR: It has been changed.
- Line 90 “the differential expression proteins” – rather “the proteins which were differenatilly expressed”
AUTHOR: It has been changed.
- Fig. 2 – “TE promotes performance”
AUTHOR: It has been deleted.
- Line 131 and 192 – “intra-articular” – rather “intraarticular”
AUTHOR: It has been changed.
- Line 139. “by bilateral medial medial meniscectomy” – “after bilateral medial medial meniscectomy?”
AUTHOR: It has been changed.
- Line 139 – “Super-139 paramagnetic iron oxide (SPIO) nanoparticles were labelled with bone marrow-derived 140 MDCs (BM-MSCs)” – rather cells were labelled with SPIOs, right?
AUTHOR: It has been changed.
- Line 146. “right side (treatment group) and the left side (control group)” – what for these right and left sides refer to?
AUTHOR: Right knee, left knee (it has been changed).
- Line 153 “all three options significantly improved early knee OA” – please elaborate on that, what exactly was improved?
AUTHOR: I have added the following paragraph:
“The combined therapy group augmented the bone volume and trabecular thickness as well as diminished synovitis more than ESWT or WJMSCs individually. However, there were no significant difference in combined ESWT and WJMSCS as demonstrated in the expressions of IGF-1 and TGF-β1 and decrease of the TUNEL activity on OA knee. Moreover, WJMSCs therapy significantly augmented the expression of the type II collagen when compared with ESWT and ESWT combined with WJMSCs in OA knee”.
- Line 196 – the explanation of the abbreviation BM-MSCs is not needed here, you should place it upwords
AUTHOR: It has been changed.
- Table 2. “papers” -rather “manuscripts”
AUTHOR: It has been changed.
YOU CAN SEE AL CHANGES DONE IN BLUE
Round 2
Reviewer 1 Report
The revised manuscript does not really address the major concerns from this reviewer. For the most part, the author provided a “shopping list” of results describing where each reference without personal opinions or thoughts which would make the review much more interesting and/or relevant to the reader. In my opinion, the author should focus on discussing some major factors that may impact the efficacy, such as allogenic vs autologous cells, primary cells vs cultured cells, differentiated vs undifferentiated cells, licensed vs unlicensed cells, the variation of cell preparations, and the clinical conditions of recipients. Because of these deficiencies, the manuscript is not suitable for publication in its present form.
Author Response
REVIEWER-1
The revised manuscript does not really address the major concerns from this reviewer. For the most part, the author provided a “shopping list” of results describing where each reference without personal opinions or thoughts which would make the review much more interesting and/or relevant to the reader. In my opinion, the author should focus on discussing some major factors that may impact the efficacy, such as allogenic vs autologous cells, primary cells vs cultured cells, differentiated vs undifferentiated cells, licensed vs unlicensed cells, the variation of cell preparations, and the clinical conditions of recipients. Because of these deficiencies, the manuscript is not suitable for publication in its present form.
AUTHOR: I have included the following paragraphs (and their corresponding References) in the DISCUSSION :
It is important to emphasize that there are some Major factors that may impact the efficacy of intraarticular injections of MSCs, such as allogenic vs autologous cells, primary cells vs cultured cells, differentiated vs undifferentiated cells, licensed vs unlicensed cells, the variation of cell preparations, and the clinical conditions of recipients
With respect to allogenic vs autologous MSCs, it has been reported by Li et al that the possible hazards and restraints of utilizing autologous vs. allogeneic MSCs in clinical practice are still deeply discussed such as the possible influence of donor–donor heterogeneity [LI-2021]. Figure 3 shows the reported advantages and disadvantages of allogeneic and autologous MSCs in the preclinical and clinical practice [LI-2021]. (FIGURE 3 – THIS IS A NEW FIGURE). 68. Li, C.; Zhao, H.; Cheng, L.; Wang, B. Allogeneic vs. autologous mesenchymal stem/stromal cells in their medication practice. Cell Biosci 2021, 11, 187.
Regarding primary MSCs vs cultured cells, Rakian et al have published that native bone marrow extracellular matrix (BM-ECM) renders a unique microenvironment that ameliorates the growth of MSCs in serum-free media (SFM) and maintains MSC quality in terms of replication, differentiation, and bone morphogenetic protein-2 (BMP-2) responsiveness. The utilization of a potent culture system consisting of native tissue specific ECM and defined SFM will permit us to prepare substantial amounts of MSCs, at the same time that keeping their stem cell characteristics, for cell-based treatments [RAKIAN-2015]. 69. Rakian, R.; Block, T. J .; Johnson, S. M.;, Marinkovic, M.; Wu, J.; Dai, Q.; Dean, D. D.; Chen, X. D. Native extracellular matrix preserves mesenchymal stem cell “stemness” and differentiation potential under serum-free culture conditions. Stem Cell Res Ther 2015, 6, 235.
In connection with differentiated MSCs vs unidiffereniated MSCs, in 2009 Zhang et al compared the outcomes of grafting into the rat contused spinal cord undifferentiated, adipose tissue-derived stromal cells (uADSCs) versus ADSCs induced by two different methods to create differentiated nervous tissue. Their findings suggested that ADSCs were able to differentiate into neural-like cells in vitro and in vivo. But, neural differentiated ADSCs did not lead to better functional recovery than undifferentiated ones [ZHANG-2009]. 70. Zhang, H. T.; Luo, J.; Sui, L. S.; Ma, X.; Yan, Z. J. ; Lin, J. H. ; Wang, Y. S. ; Chen, Y. Z. ; Jiang, X. D. ; Xu, R. X. Effects of differentiated versus undifferentiated adipose tissue-derived stromal cell grafts on functional recovery after spinal cord contusion. Cell Mol Neurobiol 2009, 29, 1283-1292.
With regard to licenced MSCs vs unlicensed MSCs, Cassano et al stated that in vitro licensing before therapeutic application could lead to a more foreseeable immunomodulatory and reparative reaction to MSC treatment compared to in vivo inflammatory licensing by the recipient environment [CASSANO]. 71. Cassano, J. M. ; Schnabel, L. V. ; Goodale, M. B. ; Fortier, L. A. Inflammatory licensed equine MSCs are chondroprotective and exhibit enhanced immunomodulation in an inflammatory environment. Stem Cell Res Ther 2018, 9, 82.
In 2020 Lynch et al rendered strong evidence for the utilization of TGF-β1 licensing as an unconventional approach for improving MSC immunosuppressive ability [LYNCH-2020]72. Lynch, K.; Treacy, O.; Chen, X.; Murphy, N.; Lohan, P.; Islam, M. N.; Donohoe, E.; Griffin, M. D.; Watson, L.; McLoughlin, S.; O'Malley, G.; Ryan, .A. E.; Ritter, T. TGF-β1-licensed murine MSCs show superior therapeutic efficacy in modulating corneal allograft immune rejection in vivo. Mol Ther 2020, 28, 2023-2043.
In 2021 Lu and Qiao stated that in spite of the heterogeneity, pre-licensing did not impact the cell-cycle and stemness of human bone marrow-derived MSCs. The osteogenesis potencies were reduced, the chondrogenesis potencies were augmented at the same time that the adipogenesis potencies were stable in licensed MSCs. Licensed MSCs also demonstrated more efficacious immunomodulate ability including expression of related chemokines, cytokines, surface molecules, and receptors [LU and QIAO]. 73. Lu, S.; Qiao, X. Single-cell profiles of human bone marrow-derived mesenchymal stromal cells after IFN-γ and TNF-α licensing. Gene 2021, 771, 145347.
With respect to the variation of cell preparation, in 2022 Duan et al affirmed that exosomes liberated by MSCs had been insinuated as good candidates for cartilage injuries and OA management, and that exosomes for clinical practice needed large-scale production. To this objective, human synovial fluid MSCs (hSF-MSCs) were grown on microcarrier beads, and then cultured in a dynamic three-dimension (3D) culture system. Through using 3D dynamic culture, this method successfully attained large-scale exosomes from SF-MSC culture supernatants. The results of this study indicated that this technique can make a great amount of good manufacturing practices (GMP)-grade exosomes. These exosomes could be used in exosome biology investigation and clinical OA management [DUAN-2022]. 74. Duan, L.; Li, X.; Xu, X.; Xu, L.; Wang, D.; Ouyang, K. ; Liang, Y. Large-scale preparation of synovial fluid mesenchymal stem cell-derived exosomes by 3D bioreactor culture. J Vis Exp 2022, Jul 26, 185.
In a study published in 2022 by Liao et al, injectable adipose-derived stem cells-embedded alginate-gelatin microspheres (Alg-Gel-ADSCs MSs) were prepared by the electrospray . Compared with the traditional alginate microspheres, its support ability for ADSCs was better and demonstrated a better repair effect. This report rendered a good approach for cartilage tissue regeneration. 75. Liao, S.; Meng, H.; Zhao, J.; Lin, W.; Liu, X.; Tian, Z.; Lan, L.; Yang, H.; Zou, Y.; Xu, Y.; Gao, X.; Lu, S.; Peng, J. Injectable adipose-derived stem cells-embedded alginate-gelatin microspheres prepared by electrospray for cartilage tissue regeneration. J Orthop Translat 2022, 33, 174-185.
Regarding the clinical conditions of recipients, Doyle et al reported that 40 × 106 MSCs were the most likely to accomplish optimal responses in patients with grade ≥ 2 knee OA. Besides, substantial ameliorations were published when utilizing inferior (24 × 106) and superior (100 × 106) cell numbers, although they caused persistent pain and inflammation. 76. Doyle, E. C.; Wragg, N. M.; Wilson, S. L. Intraarticular injection of bone marrow-derived mesenchymal stem cells enhances regeneration in knee osteoarthritis. Knee Surg Sports Traumatol Arthrosc 2020, 28, 3827-3842.
More
Xiang et al have stated that a standard protocol for intraarticular injections of MSCs in knee OA is needed. This must include the following: cell selection, authentication (phenotypic analysis and multipotent differentiation potential, particularly differentiate with progenitor cells), culture or expansion techniques, dosages, and rehabilitation program after treatment. 77. Xiang, X. N.; Zhu, S. Y.; He, H. C.; Yu, X.; Xu, Y.; He, C. Q. Mesenchymal stromal cell-based therapy for cartilage regeneration in knee osteoarthritis. Stem Cell Res Ther 2022, 13, 14

Reviewer 2 Report
Thank you for intorducing the changes, they improved your submission very nice. However, my main reservations are still valid, since you did not change the following critical issues: (or changed slightly, but not sufficiently):
- The presentation of the results. The text is now of really of good quality, but figures do not embellish the article, but rather give the impression of being hastily made. In my oppinion Fig. 1 is still not representative for the manuscript, it is definately too large, but it is informative. Maybe you could arrange it into the table or smaller figure? Changes in the Fig. 2 did not improve its quality, the colours are still aggresive, the figure is not nicely arranged into the text. Other than that, the font is bigge than in the text - please unify that. Please change that figures accordingly. I'd suggest to use a dedicated graphical program - you can download plenty of these for free from the web.
- The Tables are still to descriptive – please minimize the text in the columns: "methds and results" and "conclusions". The table shouls summarize the observations, the whole sentences are not needed, especially when a table takes 6 pages.
- Many sentences are copied word-by-word from the original publications and therefore one might suspect plagiarism – please change the text accordingly.
AUTHOR: This has been changed.
Reviewer: Well, I am not really sure. You did introduce some changes, eg. more detailed description of the models, but the main core of the descriptions has not been changed. And this connects to my another criticism:
- Sarting from the line 69 you start listing specific works and their results. This arrangement is suitable for a table rather than a survey article. In the review paper, it is necessary not only to list, but also to give a sequence (e.g. chronological or thematic) to the works cited. One must emerge from the other, and finally in science one discovery or observation mulls another. Merely describing the work done previously by others is not sufficient for a review article. Moreover, please specify in all paragraphs if the work was done on an in vitro model of OA, a cell-culture model, animal model on real patients. Maybe think of dividing the data presented here in such paragraphs.
AUTHOR: In all paragraphs I have specified the type of model used.
Reviewer: Thank you for specifying the model, but that was not the main issue of that point. What I meant is that the reader of the review article does not expect just to have a compilation of simple facts and observations. You shoul rather add youor comments and criticism and arrange tha informaation into a nice reading flow. - please make these corrections.
Author Response
REVIEWER-2
Thank you for introducing the changes, they improved your submission very nice. However, my main reservations are still valid, since you did not change the following critical issues: (or changed slightly, but not sufficiently):
- The presentation of the results. The text is now of really of good quality, but figures do not embellish the article, but rather give the impression of being hastily made. In my oppinion Fig. 1 is still not representative for the manuscript, it is definitely too large, but it is informative. Maybe you could arrange it into the table or smaller figure? Changes in the Fig. 2 did not improve its quality, the colours are still aggresive, the figure is not nicely arranged into the text. Other than that, the font is bigger than in the text - please unify that. Please change that figures accordingly. I'd suggest to use a dedicated graphical program - you can download plenty of these for free from the web.
AUTHOR: I have arranged Figure 1 into a smaller figure. Besides, I have reduced the size of figure 2 and eliminated the colors.
- The Tables are still to descriptive – please minimize the text in the columns: "methds and results" and "conclusions". The table should summarize the observations, the whole sentences are not needed, especially when a table takes 6 pages.
AUTHOR: I have minimized the columns ·methods and results” and “conclusions” a lot.
- Many sentences are copied word-by-word from the original publications and therefore one might suspect plagiarism – please change the text accordingly.
AUTHOR: I have tried to avoid plagiarism as much as possible.
Reviewer: Well, I am not really sure. You did introduce some changes, eg. more detailed description of the models, but the main core of the descriptions has not been changed. And this connects to my another criticism: Sarting from the line 69 you start listing specific works and their results. This arrangement is suitable for a table rather than a survey article. In the review paper, it is necessary not only to list, but also to give a sequence (e.g. chronological or thematic) to the works cited. One must emerge from the other, and finally in science one discovery or observation mulls another. Merely describing the work done previously by others is not sufficient for a review article. Moreover, please specify in all paragraphs if the work was done on an in vitro model of OA, a cell-culture model, animal model on real patients. Maybe think of dividing the data presented here in such paragraphs.
AUTHOR: In TABLE 1 (systematic reviews and meta-analysis) it is shown whether the reports were on patients or on animals. In TABLE 2 (devoted to RCTs in patients with osteoarthritis), IN BLUE you can see the types of MSCs used.
Reviewer: Thank you for specifying the model, but that was not the main issue of that point. What I meant is that the reader of the review article does not expect just to have a compilation of simple facts and observations. You should rather add your comments and criticism and arrange the information into a nice reading flow. - please make these corrections.
AUTHOR: I have included new paragraphs (IN RED) to add my comments and criticism.

Round 3
Reviewer 2 Report
Thank you for all the changes intorduces, now the manuscript gain a lot of the clearness and soundness. I rally appreciate the Figure 3, it is well thought. I just have one, very small comment - if you would use the same colours for the Figure 1 as well (i mean, red and blue like in Fig. 3), it would be perfect. And, is it possible to move slighlty Fig. 2 to the right? To have the left edge of the figure in the same line with the left edge of the text?
Othen than these two small technical issues, I have no other critical comments. I really like the paper now, you put a lot of work into it and it paid off.
Author Response
REVIEWER-2
Thank you for all the changes introduced, now the manuscript gain a lot of the clearness and soundness. Greatly appreciate the Figure 3, it is well thought. I just have one, very small comment - if you would use the same colours for the Figure 1 as well (i mean, red and blue like in Fig. 3), it would be perfect. And, is it possible to move slighlty Fig. 2 to the right? To have the left edge of the figure in the same line with the left edge of the text?
Other than these two small technical issues, I have no other critical comments. I really like the paper now, you put a lot of work into it and it paid off.
AUTHOR: I would like to thank the Reviewer for his/her positive comments. Following her/his suggestions, in Figure 1 I have changed the colours (I have used the same as in Figure 1), and I have moved Figure 2 slightly to the right.
